# Functional Analysis of NPC2 in Alarm Pheromone Recognition by the Red Imported Fire Ant, *Solenopsis invicta* (Formicidae: *Solenopsis*)

**DOI:** 10.3390/insects16080766

**Published:** 2025-07-25

**Authors:** Peng Lin, Jiacheng Shen, Xinyi Jiang, Fenghao Liu, Youming Hou

**Affiliations:** State Key Laboratory of Agricultural and Forestry Biosafety, Fujian Agriculture and Forestry University, Fuzhou 350002, China; m15959689635@163.com (P.L.); jc_shen_cnsh@outlook.com (J.S.); xinyijiang1116@163.com (X.J.); fenghaod@163.com (F.L.)

**Keywords:** red imported fire ant, alarm pheromone, NPC2, RNA interference, electroantennography, fluorescence competitive binding

## Abstract

Research shows that NPC2 proteins, highly expressed in insect antennae, mediate olfactory recognition of odorants. In this study, we identified SinvNPC2a as specifically and highly expressed in *Solenopsis invicta* antennae through transcriptome screening and qPCR validation. Using RNA interference, fluorescence competitive binding assays, electrophysiological tests, and behavioral analysis, we demonstrated that SinvNPC2a participates in the olfactory recognition of the alarm pheromone in worker ants.

## 1. Introduction

Insects’ olfactory systems are central to their adaptation to complex environments. Its efficient ability to recognize smells relies on a group of highly specialized olfactory proteins. These include odorant-binding proteins (OBPs), chemosensory proteins (CSPs), odorant receptors (ORs), ionotropic receptors (IRs), odorant degrading enzymes (ODEs) [1,2], and sensory neuron membrane proteins (SNMPs) [3]. Olfactory sensilla, like sensilla trichodea and sensilla basiconica, on insect antennae have tiny pores on their surface. These pores allow volatile odor molecules to enter the sensillum lymph fluid inside. Inside this fluid, odor molecules first bind to OBPs. The OBPs then carry the odor molecules to ORs located on the membranes of sensory neuron dendrites [4]. The ORs convert the chemical signal into an electrical signal. This signal is sent to the central nervous system, leading to behavioral responses [5,6]. CSPs belong to another small family of soluble proteins and have been found in several insect orders. The immunocytochemical localization of these proteins in the hemolymph of olfactory sensory organs suggests their potential involvement in chemical signal transduction in insects. In ants, CSPs are highly expressed in antennae and may function as mating recognition signals in nests [7,8]. OBPs and CSPs are key mediators for insects to sense chemical signals from the environment. Research shows that while OBPs and CSPs are mainly made of α-helices, NPC2 proteins have a β-sheet structure [9,10]. Despite this difference in structure, all three types of proteins have hydrophobic binding pockets. This suggests that NPC2 proteins might also function in binding odorants in insects [1,11].

Niemann–Pick C2 type (NPC2) proteins belong to the MD-2-related lipid recognition (ML) superfamily. Their ML domain has a structure made of multiple beta-sheets. This structure allows them to bind specifically to lipids. NPC2 proteins are involved in regulating lipid balance, triglyceride storage, and immune responses [9,12,13,14,15]. Growing evidence supports the idea that NPC2s act as a third class of OBPs. They are considered potential carriers for chemical communication. For example, NPC2s in the antennae of *Camponotus japonicus* and *Helicoverpa armigera* have been shown to bind odor molecules [16,17]. A predicted model of MmedNPC2a from a parasitic wasp suggests its key parts can bind to several cotton plant volatiles [18]. PpNPC2a from Phytoseiulus persimilis is involved in sensing plant volatiles [19]. Genomic and proteomic studies have also found NPC2s in other arthropods like ticks [20,21,22,23]. In the honeybee parasite Varroa destructor, NPC2 family members form distinct groups separate from OBPs and CSPs. This suggests that they may have different functions or have evolved differently for chemical sensing [24,25]. Although these studies show that NPC2s are present and functional in arthropod olfactory sensilla [18,19,20,21,22,23,24], their exact role in insect smell is not fully understood. In contrast, the functions of OBPs and CSPs as core parts of the insect olfactory system are well-established. Common techniques used to study OBP function, like fluorescence competitive binding assays, RNA interference, and site-directed mutagenesis, are also suitable for studying the olfactory function of insect NPC2s [26].

The red imported fire ant (RIFA), *Solenopsis invicta* Buren, is a highly destructive invasive species. Accidentally introduced to the USA in the 1930s, it has now spread to over 20 countries, including Australia and China [27,28,29]. It poses serious threats to agriculture, ecosystems, and human health [30], and causes great harm in Fujian Province [29]. As social insects, their complex behaviors, like alarm responses, are mainly controlled by pheromones. The alarm pheromone, 2-ethyl-3,6-dimethylpyrazine (EDMP), is released by worker ants when threatened. This triggers colony defense reactions [31,32]. The potential of the alarm pheromone in red imported fire ant control lies in the fact that it can effectively activate the alert state of worker ants and enhance their activity, thus significantly improving the positioning efficiency of worker ants on bait [33]. Different concentrations of EDMP may enhance alarm responses such as bait attraction and behavioral activity in red imported fire ant populations [33,34]. Research has identified SinvOBP5 as the key OBP for EDMP recognition in fire ants [26]. The olfactory and non-olfactory roles of other OBPs in this ant are also well studied [25,35,36,37,38]. However, the function of NPC2 proteins in sensing the alarm pheromone in RIFA has not been reported.

To understand the role of NPC2 in alarm pheromone recognition, this study used transcriptome data from fire ant antennae exposed to the alarm pheromone. We identified two highly expressed NPC2 genes (SinvNPC2a and SinvNPC2b) in the antennae. We analyzed their expression patterns in different tissues using quantitative PCR (qPCR). We then systematically tested their olfactory function using RNA interference (RNAi), electroantennography (EAG), insect behavior assays, and fluorescence competitive binding assays. This study helps deepen our understanding of how red imported fire ants rapidly invade and adapt to new environments. It also provides new molecular targets and a theoretical basis for developing fire ant control strategies using dsRNA delivery technology.

## 2. Materials and Methods

### 2.1. Insects Collecting and Rearing

The experimental colonies of *S. invicta* in this study were collected from the Qishan campus of Fujian Agriculture and Forestry University and Minjiang University (26° N, 119° E). Red imported fire ants were maintained in controlled artificial climate chambers under constant temperature (27 ± 1 °C), humidity (RH 75 ± 5%), and a 14:10 h light–dark photoperiod. The rearing apparatus consisted of plastic trays coated with talc powder, with daily scheduled feeding of fresh *Tenebrio molitor* larvae and defatted cotton balls saturated with sugar water. PCR detection of Gp-9 alleles confirmed that the experimental colonies exhibited monogyne social organization (Appendix A) [39].

### 2.2. Total RNA Extraction and cDNA Synthesis

Antennal samples of *S. invicta* were collected before and after treatment with alarm pheromone (1.0 μg/μL). Post-treatment specimens were immediately snap-frozen in liquid nitrogen, rapidly dissected to isolate antennal tissues, and transferred into nuclease-free tubes containing RNA later™ Stabilization Solution (AM7020, Thermo Fisher Scientific, Waltham, MA, USA). All samples were stored at −80 °C until subsequent analysis. Each sample (200 mg per treatment group) was collected before and after treatment, with three biological replicates established. Total RNA was extracted from the harvested samples using the TRIzol^®^ Reagent kit (Thermo Fisher Scientific). RNA purity and concentration were assessed via a NanoDrop™ 2000 spectrophotometer (Nanodrop, Wilmington, NC, USA), while RNA integrity was rigorously evaluated using an Agilent 2100 Bioanalyzer System (Agilent Technologies, Shanghai, China). Samples meeting quality criteria (OD260/280 ≥ 1.8; RNA Integrity Number ≥ 7.0) were subsequently subjected to transcriptome sequencing. cDNA was synthesized using the FastKing^®^ gDNA Dispelling RT SuperMix (Tiangen Biotech Co., Ltd., Beijing, China) with input RNA at 1000.0 ng/µL in 20.0 µL reactions, following the manufacturer’s protocol.

### 2.3. Differentially Expressed Genes and Phylogenetic Analysis of the NPC2 Gene

Heatmap analysis of olfactory-related gene expression profiles before and after alarm pheromone treatment was conducted using the OE-Cloud online platform (https://cloud.oebiotech.com) (accessed on 4 December 2024). Chemosensory genes were selected for hierarchical clustering heatmap analysis based on fragments per kilobase of exon per million mapped reads (FPKM) normalization. Integration of gene annotation data enabled systematic screening of differentially expressed key genes within the Niemann–Pick Type C2 (NPC2) gene family through bioinformatics interrogation. The open reading frames (ORFs) and amino acid sequences of each NPC2 gene were predicted using the ORF Finder tool (https://www.ncbi.nlm.nih.gov/orffinder/) (accessed on 1 December 2024). Protein molecular weights and isoelectric points were computed via ExPASy (http://www.ExPASy.org) (accessed on 1 December 2024). Conserved domains were analyzed using the NCBI CD-Search tool (https://www.ncbi.nlm.nih.gov/Structure/cdd/wrpsb.cgi) (accessed on 6 December 2024). Signal peptides were predicted through the Department of Health Technology (https://services.healthtech.dtu.dk/services/SignalP-6.0/) (accessed on 6 December 2024). NPC2 gene-related sequences were retrieved from NCBI and subjected to multiple sequence alignment analysis using Clustal W (https://www.genome.jp/tools-bin/clustalw) (accessed on 8 December 2024). A phylogenetic tree was constructed using the maximum likelihood method in MEGA 7.0, with node values representing bootstrap percentages based on 1000 replicates.

### 2.4. Relative Expression Level of SinvNPC2

To investigate the tissue-specific expression profiles of NPC2 genes in *S. invicta*, 100 adult workers were dissected to collect antennae, heads, thoraxes, and abdomens. Four biological replicates were prepared for each tissue type. All samples were preserved in RNA later™ Stabilization Solution (AM7020, Thermo Fisher Scientific, Waltham, MA, USA) and stored at −80 °C for subsequent analysis. Total RNA extraction and cDNA synthesis were performed following the aforementioned procedures. q-PCR was conducted using Magen Universal SYBR qPCR Mix (MD70101, Magen Biotechnology, Guangzhou, China) with the following reaction system: 10.0 µL SYBR qPCR Mix, 1 µL forward primer, 1.0 µL reverse primer, 1.0 µL cDNA template, and 7.0 µL RNase-free water. Thermal cycling conditions included 95 °C for 30 s (initial denaturation), followed by 40 cycles of 95 °C for 30 s (denaturation) and 60 °C for 30 s (annealing/extension). The comparative Ct method (2^−ΔΔCt^) was employed to normalize gene expression levels, using elongation factor 1-beta (ef1-β, GenBank accession: EH413796) as the endogenous reference [40]. Three biological replicates and four technical replicates were analyzed for each sample. Primer sequences for qPCR are provided in Appendix A.

### 2.5. RNA Interference

Double-stranded RNAs (dsRNAs) targeting SinvNPC2 genes were synthesized using gene-specific primers containing T7 RNA polymerase promoter sequences (Appendix A). Sequence-verified plasmids containing the target regions served as PCR templates (Appendix A). dsRNA synthesis and purification were performed using the T7 RiboMAX™ Express RNAi System (Promega, Madison, WI, USA), following the manufacturer’s protocol. Purified dsRNAs were quality-controlled through dual verification: concentration measurement using a NanoVue™ spectrophotometer (GE Healthcare, Wuxi, China) and integrity assessment via 1% agarose gel electrophoresis. For RNA interference experiments, three experimental groups (dsSinvNPC2a, dsSinvNPC2b, dsEGFP control) were established, each containing three biological replicates (30 workers per replicate). Each worker received 1.0 µL of 2500 ng/µL dsRNA solution through oral administration. Treated ants were collected at 24, 36, and 48 h post-treatment for qPCR analysis. Gene expression quantification followed the previously described 2^−ΔΔCt^ method.

### 2.6. Electroantennography

Electroantennogram (EAG) analysis was performed on randomly selected worker ants from control groups (dsEGFP and water-fed) and 24 h RNAi-treated groups. Antennal responses to 2-ethyl-3,6-dimethylpyrazine (EDMP) were tested at five concentrations: 0.01, 0.1, 1.0, 10.0, and 100.0 µg/µL, The appropriate concentration was selected according to the method of Di Guan et al. [33]. Antennae-intact heads were dissected using microsurgical blades and affixed to Syntech PR-05 dual electrodes with SignaGel^®^ Electrode Gel (Parker Laboratories, Inc., Fairfield, NJ, USA). Filter strips (0.4 cm × 0.5 cm) were adhered to the inner wall of 1 mL polypropylene pipette tips. Test samples (20.0 µL) were loaded at the center of filter strips, maintaining ≥ 6 mm between the filter apex and tip opening to ensure stable airflow. Antennae were positioned 1 cm from the stimulus delivery port. Purified airflow (50.0 cm/s) was controlled by a Syntech CS-55 stimulus controller (Syntech GmbH, Kirchzarten, Germany). Stimulus parameters included 0.5 s pulse duration and ≥60 s inter-stimulus intervals to minimize adaptation. Biopotential signals were amplified via a Syntech IDAC-4 amplifier and recorded using EAGPro software (v.2.0.2). Peak amplitudes were measured under quintuplicate.

### 2.7. Behavioral Assays

A DeepLabCut-based behavioral quantification model was established, with the training dataset undergoing 100,000 iterations. Model performance metrics included training error (2.56 pixels) and testing error (3.32 pixels), with a confidence threshold set at 0.6 [41]. The validated model was subsequently applied to analyze all experimental videos, followed by Python v.2.6 script-driven behavioral pattern classification. Ants from control (untreated) and 24 h RNAi-treated groups were subjected to EDMP exposure at three concentrations (0.1, 1.0, and 10.0 µg/µL), selected based on the results of EAG response and Li, Y.-Y. et al. [35]. Behavioral assays were conducted in polystyrene Petri dishes (9 cm diameter) coated with talcum powder along the periphery. The lid was radially divided into five 1.8 cm-wide sectors centered on the dish axis (Appendix A) (Region 1 is the ant colony gathering area; Region 2 is the transition zone adjacent to the center; Region 3 serves as the intermediate zone and the alarm pheromone release zone; Region 4 is the peripheral zone; Region 5 is the most marginal zone). Sector 1 served as the colony aggregation zone, pre-treated with 10 µL honeydew to stabilize worker clusters. After colony stabilization, 20 µL alarm pheromone solution was applied to a filter paper disk positioned at the dish center for stimulation. A total of 50 worker ants from control and RNAi-treated groups (five replicates per group) were exposed to three alarm pheromone concentrations (0.1, 1.0, and 10.0 µg/µL). The total number of ants evacuating Sector 1 within 5 min (15 observational timepoints at 20 s intervals) and spatial redistribution percentages across adjacent sectors (2–5) were quantified.

### 2.8. Recombinant Protein Expression and Purification of SinvNPC2

The pET28a plasmid carrying SinvNPC2a was transformed into *E. coli* BL21 (DE3) pLysS competent cells for recombinant protein expression. A single colony was inoculated into 3.0 mL LB medium containing 50 μg/mL ampicillin and cultured overnight at 37 °C with 220 rpm shaking. The culture was then diluted 1:100 into 30 mL fresh LB medium with 50.0 μg/mL ampicillin and grown to OD600 0.6–0.8 under identical conditions. Protein induction was initiated by adding IPTG (0.5 mM final concentration) followed by 4 h induction at 37 °C with 220 rpm shaking and Overnight induction at 11 °C with 220 rpm shaking. Cells were pelleted by centrifugation (4000× *g*, 10 min), resuspended in PBS, and lysed via sonication. Soluble (supernatant) and insoluble (pellet) fractions were analyzed by 12% SDS-PAGE with Coomassie Blue staining. For protein purification, the supernatant was loaded onto a Ni-IDA Sepharose CL-6B column pre-equilibrated with binding buffer (20 mM Tris-HCl, 0.5 M NaCl, pH 8.0) at 0.5 mL/min. The purified protein was dialyzed against PBS under 4 °C overnight, confirmed by SDS-PAGE, and stored at −80 °C for fluorescence competitive binding assays.

### 2.9. In Vitro Binding Assays

Fluorescence competitive binding assays (FCBA) were conducted to investigate the binding properties of SinvNPC2a with 2-ethyl-3,6-dimethylpyrazine (EDMP). EDMP (purity > 95%) was purchased from Shanghai Macklin Biochemical Co., Ltd. (Shanghai, China). All assays were performed using a BioTek Synergy H1 Multiscan Spectrum system. For 1-NPN (N-phenyl-1-naphthylamine) affinity measurements, purified SinvNPC2a (2.0 μmol/L) was titrated with 1 mmol/L 1-NPN to achieve final concentrations ranging from 4.0 to 32.0 μmol/L. Fluorescence excitation was set at 337 nm, and emission spectra were recorded between 337 and 500 nm. Both emission and excitation slits were fixed at 10 nm. To determine EDMP binding affinity, a mixture containing SinvNPC2 (2.0 μmol/L) and 1-NPN (2.0 μmol/L) was titrated with 1 mmol/L EDMP to final concentrations of 4–20 μmol/L. The inhibition constant (*K*_i_) was calculated using the formula *K*_i_ = IC_50_/ (1 + [1-NPN]/*K*_1-NPN_), where IC50 represents the EDMP concentration required to reduce 1-NPN fluorescence intensity by 50%, [1-NPN] denotes the free 1-NPN concentration, and 1-NPN is the dissociation constant of the SinvNPC2a/1-NPN complex. *K*_i_ < 5.0 μM (strong binding ability), 5.0 μM ≤ *K*_i_ < 10.0 μM (moderate binding ability), and *K*_i_ ≥ 10.0 μM (weak binding ability), according to the method of Huang et al., 2023 [42].

### 2.10. Structural Modeling and Molecular Docking

The three-dimensional structure of SinvNPC2a was predicted using AlphaFold 3.0 (alphafoldserver.com/) (accessed on 26 December 2024). This artificial intelligence system enabled template-free modeling and generated a high-precision protein structure. Model quality assessment was performed through two complementary approaches: (1) the built-in AlphaFold evaluation metric (predicted Local Distance Difference Test, pLDDT), providing residue-level confidence scores (0–100 scale), and (2) three established validation tools—ERRAT, VERIFY 3D, and PROCHECK (https://saves.mbi.ucla.edu/) (accessed on 3 June 2025). For molecular docking analysis between NPC2a and EDMP, the CB-DOCK 2 web server (https://cadd.labshare.cn/cb-dock2/php/index.php) (accessed on 3 June 2025) was employed to investigate potential binding interactions. Finally, the result of molecular docking was visualized using PyMOL 2.6 software.

### 2.11. Statistical Analysis

Tissue-specific expression of SinvNPC2a was analyzed by qPCR using the least significant difference (LSD) test. Temporal differences in gene expression before and after RNAi treatment were assessed via independent samples t-test. Electrophysiological (EAG) and behavioral trajectory data were subjected to Tukey’s honestly significant difference (HSD) test, with * *p* < 0.05 indicating statistical significance. All statistical analyses were performed in SPSS Statistics 23.0 (IBM Corp., Armonk, NY, USA), with graphical representations generated using GraphPad Prism 9.0 (GraphPad Software, San Diego, CA, USA).

## 3. Results

### 3.1. Validation of RIFA Antennal Transcriptome Data

After screening the transcriptome data of red fire ant antennae before and after EDMP treatment, the differential genes of olfactory pathway were identified (Appendix A), and the cluster analysis of these genes revealed that there were two highly expressed and different SinvNPC2 genes (Appendix A). SinvNPC2a and SinvNPC2b both possess an ORF, encoding 462 base pairs (bp) and 468 bp, respectively. Their theoretical molecular weights are 17.03 kDa and 16.88 kDa, with isoelectric points (pI) of 5.09 and 5.17 (Table 1). An ML conserved domain was identified in both SinvNPC2 proteins (Figure 1A). Signal peptide prediction indicated that both proteins contain a signal peptide, with lengths ranging between 18 and 20 amino acids (Figure 1A). Based on transcriptome screening, multiple sequence alignment of the amino acid sequences of SinvNPC2a, SinvNPC2b, and other Hymenopteran NPC2s (accession numbers for sequences used in this alignment are listed in Appendix A) revealed that SinvNPC2a contains six conserved cysteine (Cys) residues, consistent with NPC2 protein sequences from nine other species examined. In contrast, SinvNPC2b has only five conserved Cys residues (Figure 1B).

### 3.2. Identification and Analysis of SinvNPC2 Genes

To further investigate the phylogenetic relationships of SinvNPC2 proteins, we performed a BLASTp analysis against the NCBI database to identify homologous sequences. A phylogenetic tree was constructed using amino acid sequences of NPC2 proteins from RIFA (SinvNPC2a and SinvNPC2b), 30 species with high homology (belonging to Hymenoptera, Coleoptera, Diptera, and Hemiptera), and NPC2 proteins previously documented to function in chemoreception. The resulting phylogenetic tree revealed distinct clustering: SinvNPC2a grouped within a chemosensory-functional clade alongside CjapNPC2, MmedNPC2a, HarmNPC2-1, and BdioNPC2b, proteins confirmed to participate in semiochemical binding (Figure 2). In contrast, SinvNPC2b clustered within a separate branch (Figure 2).

### 3.3. Relative Transcript Level of SinvNPC2 Genes

Relative transcript profiling revealed that both SinvNPC2a exhibit antennal-specific expression (*p* < 0.001) (Figure 3). Expression levels of SinvNPC2a in the head, thorax, and abdomen were negligible. Based on this pronounced antennal-specific expression pattern, we hypothesize that SinvNPC2a, analogous to OBP5 [26], may function in the olfactory recognition of pheromone.

### 3.4. RNA Interference Assays

We employed RNA interference (RNAi) to achieve targeted knockdown of both genes. The knockdown efficiency was assessed using quantitative real-time PCR (qPCR), which quantified the mRNA levels of SinvNPC2a and SinvNPC2b in the antennae of ants injected with dsRNA targeting either SinvNPC2a (dsSinvNPC2a) or SinvNPC2b (dsSinvNPC2b), or with the CK (dsGFP). The results demonstrated a significant suppression of SinvNPC2a expression in RIFA injected with dsSinvNPC2a compared to the CK group. Specifically, expression levels were significantly reduced by 91.52% at 24 h (*p* < 0.05), 92.76% at 36 h (*p* < 0.01), and 97.91% at 48 h (*p* < 0.001) (Figure 4).

### 3.5. EAG and Behavioral Responses of RIFA

To validate whether EDMP could elicit an EAG response in RIFA following RNAi knockdown, this study employed five concentrations of EDMP as olfactory stimuli. Compared to worker ants injected with dsEGFP, those injected with dsSinvNPC2a exhibited significantly attenuated EAG responses to EDMP (at 0.1 µg/µL: *p* < 0.05; at 100.0 µg/µL: *p* < 0.05). Specifically, at concentrations of 0.1, 1.0, and 10.0 µg/µL, the EAG responses to EDMP in the dsSinvNPC2a-treated group were significantly reduced compared to the dsEGFP control group (*p* < 0.01). In contrast, EAG responses to the CK (*n*-hexane) showed no significant differences (Figure 5). Following SinvNPC2a knockdown, behavioral trajectory imaging revealed a significant decrease in worker ants’ locomotor activity. The activity range, trajectory coverage area, and movement speed were all significantly reduced compared to the dsEGFP control group (Figure 6A,B). Behavioral assays demonstrated that the number of worker ants escaping from Zone I was significantly lower in the dsSinvNPC2a-treated group compared to the control (Figure 6C). Worker distribution in response to EDMP also differed. At 0.1 µg/µL EDMP, distribution in Zones III, IV, and V showed no significant difference between treated and control groups (*p* > 0.05), but a significant difference was observed in Zone II (*p* < 0.05). At higher concentrations (1.0 µg/µL and 10.0 µg/µL), significant differences in worker ants’ distribution were found in Zones II, III, and IV (*p* < 0.05) compared to CK (Figure 6D).

### 3.6. Molecular Docking

SinvNPC2a was used as the target protein for 3D model construction, and the EDMP were docked with SinvNPC2a. A 3D model of SinvNPC2a (Appendix A) was constructed using the AlphaFold 3.0 online program. The quality of this model was subsequently assessed using pLDDT, ERRAT, and PROCHECK. The results indicated high credibility for the 3D model of SinvNPC2a constructed with AlphaFold 3.0 without a template (Appendix A). The molecular docking results show that EDMP may bind tightly to SinvNPC2a, with a *Δ*G_binding_ of −4.8 kcal/mol (Figure 7; Appendix A).

### 3.7. In Vitro Binding Assays

Recombinant protein purity and concentration were assessed via SDS-PAGE. Purified recombinant SinvNPC2a protein was analyzed using SDS-PAGE to confirm purity and determine concentration (Appendix A). The dissociation constant (*K*_d_) was determined. Experimental results revealed *K*_d_ of 15.59 μM for SinvNPC2a (Table 2). In addition, the Scathard equation analysis showed that SinvNPC2a and 1-NPN could bind to each other in a 1:1 ratio and showed a linear relationship (R^2^ > 0.95) (Figure 8A), which could be used as a suitability probe for subsequent experiments. The ability of EDMP to competitively bind to SinvNPC2a was tested using the fluorescence competitive binding assay (FCBA). The competitive binding standard curve is presented in Figure 8A, and the corresponding binding curve for SinvNPC2a is shown in Figure 8B. Results from the FCBA with the EDMP against SinvNPC2a indicated high binding affinity for EDMP.

## 4. Discussion

This study provides the first clear evidence that a specific protein called SinvNPC2a, found only in the antennae of fire ants, plays a key role in helping worker ants detect EDMP—the main chemical in their alarm signal.

Our findings strongly support the hypothesis that NPC2 proteins function as olfactory proteins in insects alongside OBPs and CSPs, participating in chemical communication [1,16]. Sequence comparison and evolutionary analysis revealed that SinvNPC2a clusters together with other NPC2 proteins known to sense chemicals. These include CjapNPC2, which regulates ant social behaviors [16]; MmedNPC2a, which detects plant volatiles [18]; HarmNPC2-1, which binds gossypol [17]; and BdioNPC2b, which binds aldehydes [43]. SinvNPC2a also shares critical conserved structural features with them, such as six cysteine residues. Furthermore, our in vivo functional blocking experiments and in vitro binding tests provide direct evidence for NPC2 function in pheromone recognition, a specific form of chemical communication. SinvNPC2a represents the first identified NPC2 protein known to recognize a pyrazine alarm pheromone. This significantly expands our understanding of the diversity of chemicals that NPC2 proteins can interact with and the functional contexts in which they operate. Notably, while SinvNPC2b is also expressed in antennae, it lacks one conserved cysteine residue. Our RNAi approach failed to effectively reduce its expression level, possibly due to off-target effects (Appendix A) [44], suggesting its function may differ from SinvNPC2a. SinvNPC2b potentially serves a non-olfactory role or recognizes different chemical signals, and its specific functions need to be further investigated in the future. This finding provides further evidence for significant functional variation among NPC2 proteins within the same arthropod species [16,20,45].

For fire ants, fast group defense or escape triggered by alarm pheromones is vital for their survival as a social insect colony [46,47,48]. In this study, disrupting SinvNPC2a did two key things. First, it weakened the electrical nerve response of individual worker ants to EDMP. More importantly, it blocked their natural group behavior reactions to EDMP when tested in a colony setting. These affected behaviors included reduced movement, limited spreading out, and changes in escape actions. These results show that losing SinvNPC2a breaks down the entire smell-response pathway. It disrupts how the initial smell detection leads to coordinated group behavior, weakening the colony’s overall alarm response ability. Interestingly, earlier research found that another protein, SinvOBP5, is also crucial for fire ant alarm communication using pheromones [26]. Based on this finding and our new results, the highly expressed SinvNPC2a protein in fire ants’ antennae may complement OBPs in alarm pheromone recognition. We hypothesize that SinvNPC2a and SinvOBP5 might work together, or separately, to recognize or carry EDMP. We think they might interact in the same detection pathway. Alternatively, they might independently detect different forms or amounts of the same pheromone. Another possibility is that they handle different, but related, tasks in the smell process. These hypotheses need to be further verified by immunofluorescence co-localization and transcriptome joint analysis in the future [49,50]. Understanding exactly how SinvNPC2a interacts with OBPs and other known smell proteins like CYP450 enzymes within the antennae fluid [51], and how they work together over time and space, is the essential next step to fully explain the complete pheromone detection system in fire ants. Although our lab binding experiments clearly showed that SinvNPC2a physically interacts with EDMP, more work is needed. Specifically, experiments using specially mutated proteins are required to identify the precise locations where SinvNPC2a connects to EDMP. Identifying these key contact sites is fundamental to understanding the structure of its binding pocket and the detailed molecular mechanism it uses to recognize the pheromone.

In this study, we successfully reduced SinvNPC2a activity using a feeding method with dsRNA. However, this same method did not significantly reduce SinvNPC2b levels. The specific function of SinvNPC2b therefore requires further investigation using more efficient delivery methods. Secondly, the concentrations of EDMP used in our behavioral tests might be higher than those typically found under natural conditions. Future research needs to test worker ant behavioral responses using concentrations closer to natural levels and predict more accurately the behavior and ecological relationships of insects in their natural environment. While our RNAi experiments and behavioral tests confirm that SinvNPC2a function is essential, the precise way it works within the sensory structures of the antenna still needs detailed exploration.

This work significantly deepens our understanding of how social insects detect smells at the molecular level. Importantly, it reveals the key role NPC2 proteins play in pheromone-triggered alarm behavior. Furthermore, it provides strong evidence supporting the evolutionary shift in NPC2 function from transporting fats in vertebrates to supporting chemical communication systems in arthropods. This study identifies SinvNPC2a as a core molecular target for fire ant alarm behavior. Because alarm behavior is fundamental to fire ant colony defense and survival, developing RNAi-based control strategies targeting SinvNPC2a shows significant potential [44,52,53,54]. Such strategies aim to specifically disrupt their chemical communication network. Compared to traditional insecticides, this approach could be more environmentally friendly and specific to fire ants. Future work will focus on developing better systems to deliver SinvNPC2a dsRNA and combining chemical ecology and group behavior (corpse carrying mechanisms) to design multidimensional prevention and control strategies [55,56]. Success in this area will provide a solid foundation for new, environmentally friendly control technologies targeting fire ants.

## Figures and Tables

**Figure 1 insects-16-00766-f001:**
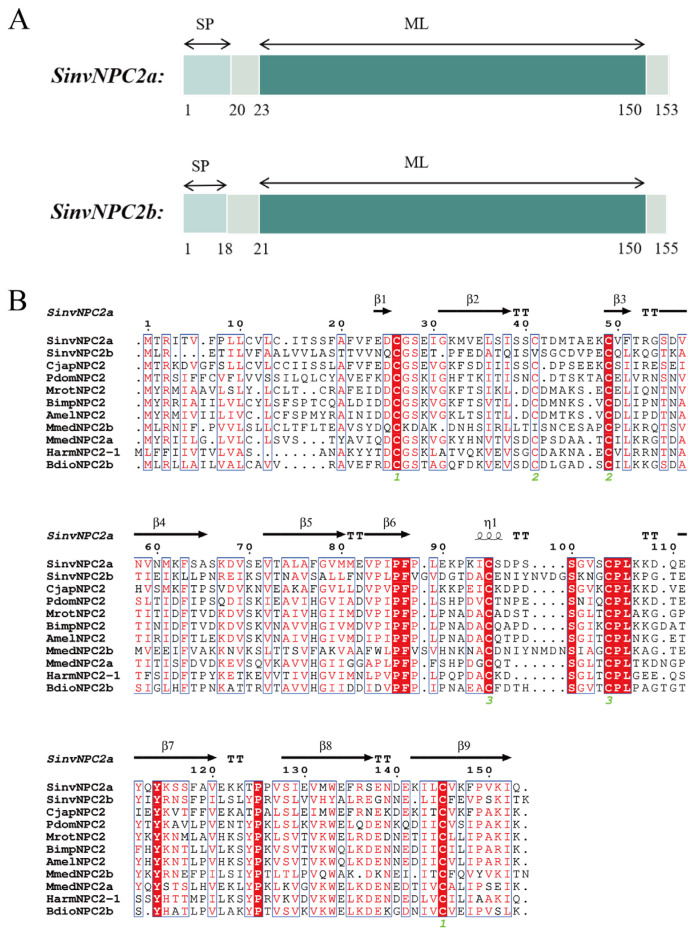
Sequence structure and sequence comparison analysis of SinvNPC2. (**A**) A schematic comparison of the domains of SinvNPC2a and SinvNPC2b. SP represents the signal peptide transported by the Sec transposon. (**B**) Sequence alignment analysis of SinvNPC2a, SinvNPC2b, and the NPC2s from *Camponotus japonicus*, *Polistes dominula*, *Megachile rotundata*, *Bombus impatiens*, *Apis mellifera*, *Microplitis mediator*, *Helicoverpa armigera*, and *Phymastichus coffea*. The red regions indicate conserved areas and the protein structure database analysis shows that SinvNPC2a has a β-fold structure compared to other species.

**Figure 2 insects-16-00766-f002:**
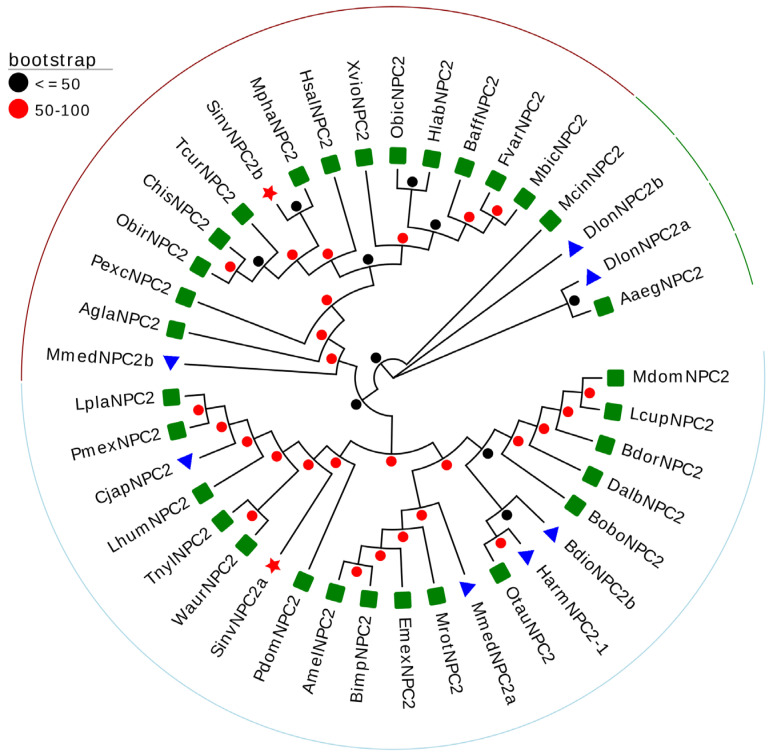
Phylogenetic relationship between SinvNPC and other species NPC2s. The phylogenetic analysis of SinvNPC2a, SinvNPC2b, and other insect NPC2s (red pentagrams represent target genes, blue triangles indicate NPC2s previously shown to bind chemicals in previous studies, green squares indicate NPC2s with Blastp homologous sequences). The phylogenetic tree was constructed using the maximum likelihood method, with values representing percentages based on 1000 repetitions. The NPC2s used for phylogenetic analysis are listed in Appendix A.

**Figure 3 insects-16-00766-f003:**
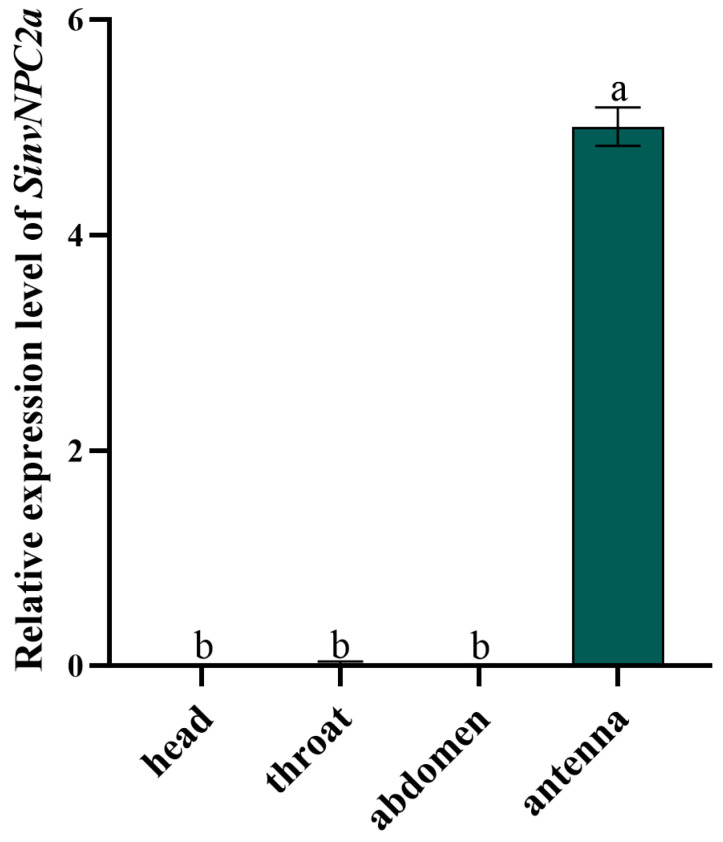
The tissue-specific expression profile of SinvNPC2a. The mRNA levels of SinvNPC2a in different tissues were analyzed using qPCR. In the qPCR analysis, mRNA levels were normalized to ef1-β levels, with data from three biological replicates, each including four technical replicates. The bars and error bars represent the mean ± standard deviation (n = 4). Different letters on the bars indicate significant differences between tissues, as determined by Tukey’s HSD test (one-way ANOVA, *p* < 0.05).

**Figure 4 insects-16-00766-f004:**
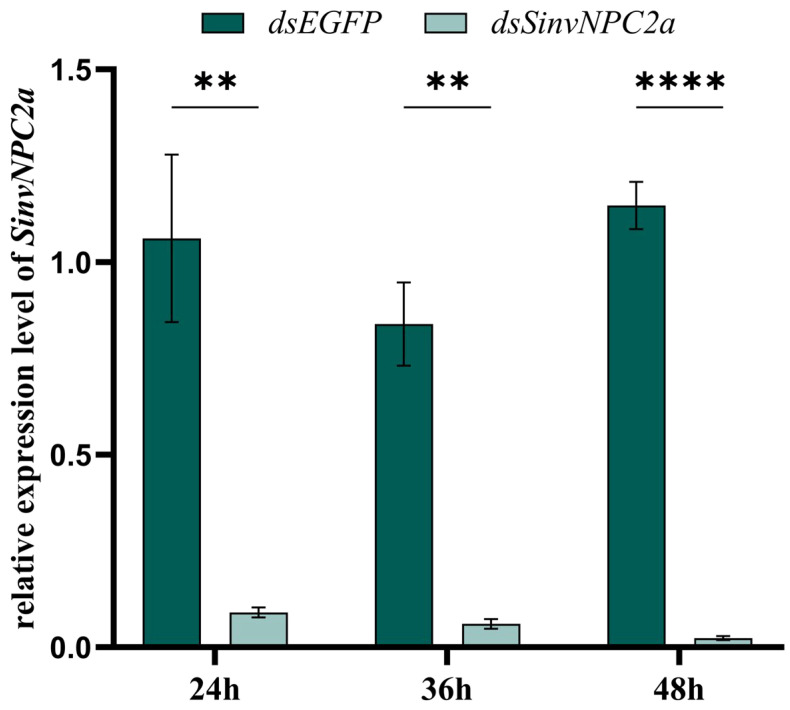
The interference efficiency of SinvNPC2a after feeding with gene-specific dsRNA. Expression of SinvNPC2a after feeding with dsSinvNPC2a. The expression levels of SinvNPC2s transcripts were analyzed using qPCR. The data shown are the mean ± SD from three biological replicates, with asterisks indicating significant differences detected by two-tailed *t*-tests (** *p* < 0.01, **** *p* < 0.001).

**Figure 5 insects-16-00766-f005:**
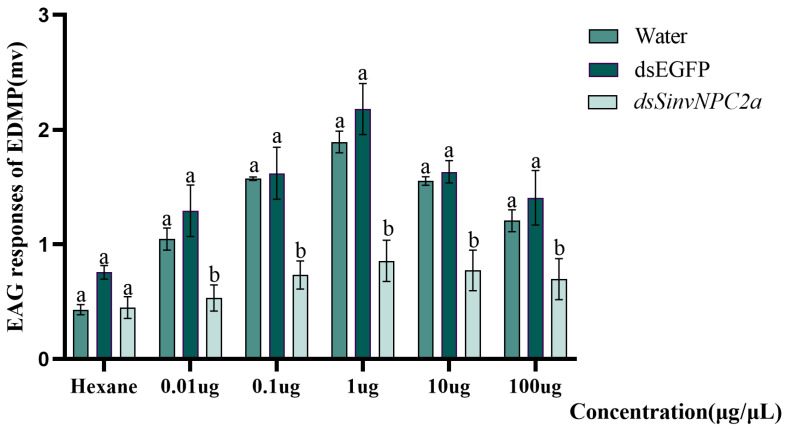
Worker ants’ response to different concentrations of EDMP (0.01, 0.1, 1.0, 10.0, and 100.0 µg/µL) after 36 h of feeding with dsSinvNPC2a. The bars and error bars represent the mean ± standard deviation (n = 3), and Tukey’s HSD test was used for one-way ANOVA, and different letters indicated that the difference between different treatments was significant, with *p* < 0.05.

**Figure 6 insects-16-00766-f006:**
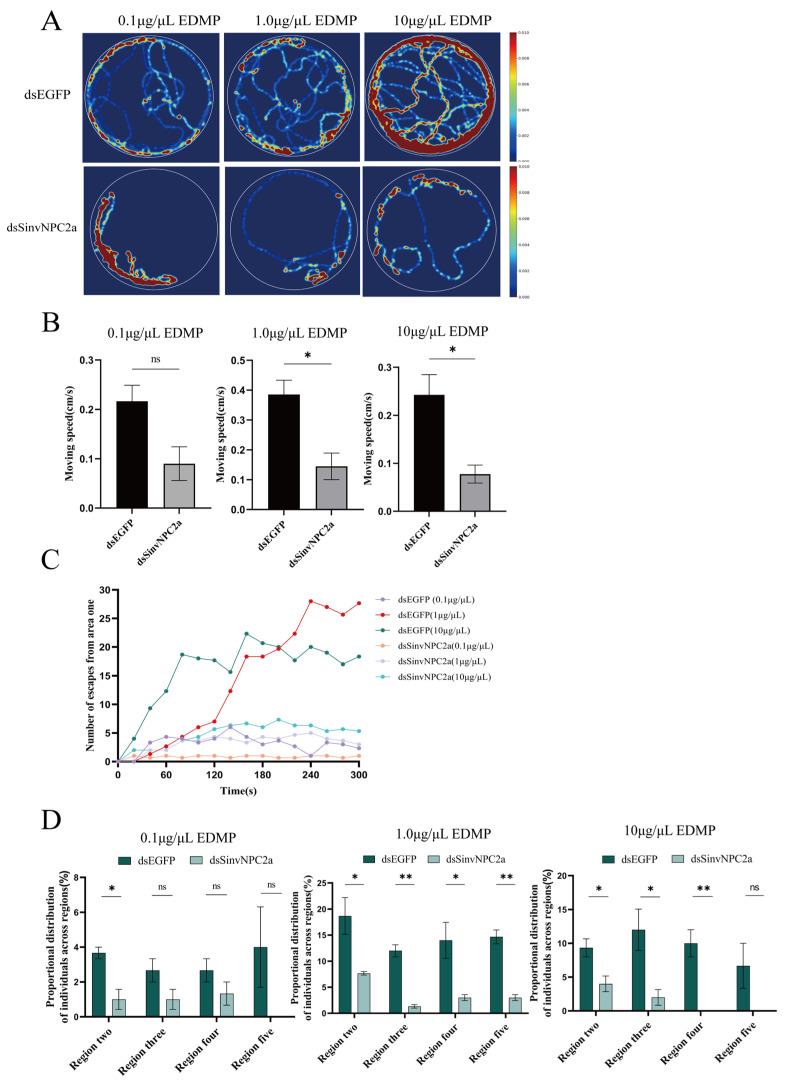
The behavioral responses of worker ants to different concentrations of EDMP after 36 h of feeding with dsEGFP and dsSinvNPC2a. (**A**) Thermal images showing the trajectories of worker ants responding to 0.1, 1.0, and 10.0 µg/µL EDMP. (**B**) The movement speed responses of worker ants to 0.1, 1.0, and 10.0 µg/µL EDMP. (**C**) The number of worker ants that spread to non-region one under 0.1, 1.0, and 10.0 µg/µL EDMP treatments (recorded every 20 s for five minutes). (**D**) The percentage of ants spreading to each region under 0.1, 1.0, and 10.0 µg/µL EDMP treatments. The bars and error bars represent the mean ± standard deviation (n = 3), with stars (*), indicating significant differences (ns, *p* > 0.05, * *p* < 0.05, ** *p* < 0.01, Student’s *t*-test).

**Figure 7 insects-16-00766-f007:**
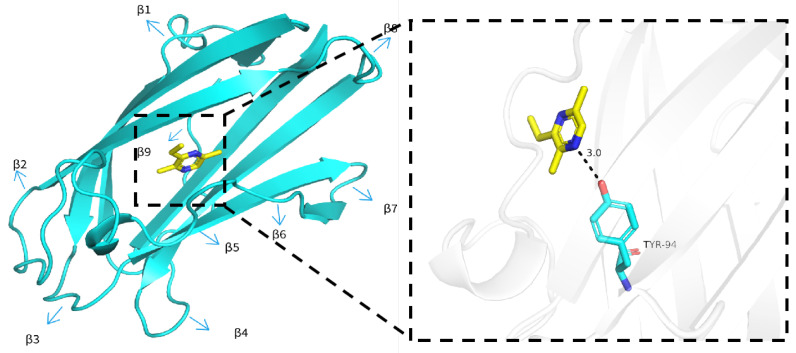
SinvNPC2a protein docking with EDMP.

**Figure 8 insects-16-00766-f008:**
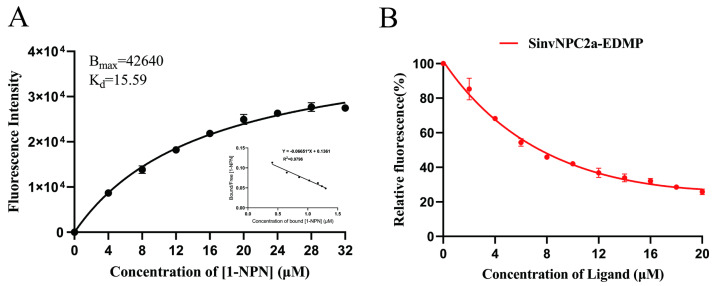
Ligand binding assays of SinvNPC2a with *n*-phenyl-1-naphthylamine (1-NPN) and EDMP at pH 7.4. Bmax represents the maximum fluorescence value. (**A**) Fluorescence competitive binding curve of SinvNPC2a with 1-NPN. (**B**) Binding curve of SinvNPC2a with EDMP.

**Table 1 insects-16-00766-t001:** The properties of the NPC2 that are highly expressed in the antennae of *S. invicta*.

GenBank Accession Number	Gene Name	Amino Acid	ORF Full Length (bp)	MW (KDa)	PI
XP_011161897.1	SinvNPC2a	153	462	58.398	8.99
XP_011170763.1	SinvNPC2b	155	468	58.941	6.92

Note: MW represents molecular weight; PI represents isoelectric point.

**Table 2 insects-16-00766-t002:** Binding affinities of SinvNPC2a for the tested ligands.

Compounds	CAS Number	IC_50_ (μM)	*K*_i_ (μM)
2-ethyl-3,6-dimethylpyrazine	27043-05-6	9.82	5.09 ± 1.51

## Data Availability

Data are contained within the article and Appendix A.

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
