# Peer review of "Functional Analysis of NPC2 in Alarm Pheromone Recognition by the Red Imported Fire Ant, Solenopsis invicta (Formicidae: Solenopsis)"

_insects, 2025, doi:10.3390/insects16080766_

Round 1

Reviewer 1 Report

Comments and Suggestions for Authors

The manuscript titled “Functional Analysis of Niemann-Pick Type C2 (NPC2) in Alarm Pheromone Recognition by the Red Imported Fire Ant, Solenopsis invicta” presents a well-structured and comprehensive study on the role of SinvNPC2a in alarm pheromone detection. The authors utilized transcriptome screening, RNAi, electrophysiological assays (EAG), behavioral tracking, in vitro binding assays, and molecular modeling to demonstrate that SinvNPC2a plays a key role in EDMP recognition. The study addresses an important question in insect chemical ecology and offers potential for novel pest control strategies. However, there are several areas that require revision and clarification.

Major Suggestions

Functional Validation of SinvNPC2b is Incomplete

Although both SinvNPC2a and SinvNPC2b were identified as highly expressed in antennae, the functional role of SinvNPC2b remains unexplored due to ineffective RNAi knockdown.

Please discuss possible reasons for this failure (e.g., delivery route, sequence design, tissue-specific expression).

If additional experiments are not feasible, clarify this limitation and suggest future research directions.

Lack of Ecological Context for EDMP Concentrations

The concentrations of EDMP used in behavioral assays (0.1–10.0 μg/μL) are relatively high. However, the relevance of these concentrations to natural conditions is unclear.

Please justify the choice of concentrations with references, or discuss the potential limitations if natural pheromone levels are unknown.

Speculative Discussion of NPC2–OBP Interaction

The manuscript suggests potential cooperation between SinvNPC2a and OBP5 in EDMP recognition, but no data directly support this.

Please make it clear that this is a hypothesis. Adding a conceptual figure to illustrate the proposed detection pathway may improve clarity.

If possible, mention any co-expression evidence, or propose how this could be tested in future studies.

Improve Introduction Structure and Focus

The Introduction is informative but could benefit from clearer structure and less repetition.

Please consider organizing it in a logical progression: insect olfaction → roles of OBPs/CSPs → emerging evidence for NPC2 in olfaction → study rationale.

Minor Suggestions

Language and Clarity

The manuscript contains some grammatical and stylistic issues.

Please revise the text for clarity, academic tone, and fluent English expression. Examples include:

“These results provide the first clear evidence...” → “These findings provide direct evidence...”

“NPC2 proteins... work together with OBPs to recognize the alarm pheromone.” → “NPC2 proteins may complement OBPs in alarm pheromone recognition.”

Figures and Legends

Some figures lack sufficient explanation, and statistical annotations are inconsistent.

Clearly define regions (e.g., Zone I–V in behavioral assays).

Standardize use of asterisks (e.g., *, **, ***), and include corresponding p-values in figure legends.

Binding Affinity Interpretation

The criteria used to categorize binding strength based on Ki values (e.g., <5 μM = strong) should be explicitly explained in the Methods section with a reference.

This study provides valuable insight into the olfactory mechanisms of fire ants and highlights SinvNPC2a as a promising molecular target for RNAi-based control strategies. With revisions to clarify experimental limitations, improve structural coherence, and strengthen the discussion, this manuscript will make a strong contribution to the field.

Author Response

Thank you very much for the valuable feedback provided on our study. Over the course of the past three days, we have extensively and carefully revised the manuscript, taking your suggestions into account.We have marked the revisions in the manuscript and indicated them in red, as shown in the attachment.In our detailed responses, provided separately, we have addressed each reviewer's comments individually. We greatly appreciate your consideration of our revised manuscript and look forward to receiving the final decision from the editor and reviewers.

Reviewer 2 Report

Comments and Suggestions for Authors

In this MS, the functional analysis of NPC2 in alarm pheromone recognition by the red imported fire ant were studied and it was demonstrated that SinvNPC2a participates in the olfactory recognition of alarm pheromone in worker ants by using RNA interference, fluorescence competitive binding assays, electrophysiological tests and behavioral analysis. The topic is very interesting and ovel study. While there are some shortcomings which are follwong as:

 Q1: The title should be change as “Functional Analysis of NPC2 in Alarm Pheromone Recognition by the Red Imported Fire Ant Solenopsis invicta”.

 Q2: Introduction: “Our results show that SinvNPC2 proteins, highly expressed in fire ant antennae, work together with OBPs to recognize the alarm pheromone. Disrupting the function of SinvNPC2a could offer a new approach for controlling fire ant spread." All thses are results, they should be deleted from the section of Introduction.

 Q3: 2.1. Insects collecting and rearing: What about the PCR dection of Gp-9 alleles confirmation? Give the results here!

 Q4: Figure 1 and Figure 2: No NPC2s were found in the figure B.

 Q5: Figure 3: For RNA interference experiments, three experimental groups (dsSinvNPC2a, dsSinvNPC2b, dsEGFP control) were established. In this figure, what about the SinvNPC2s? The results of SinvNPC2a and SinvNPC2b should be given respectively.

 Q6: Figure 4: In Figure 4A, not dsSinvNPCa, it is dsSinvNPC2a.

 Q7: Figure 5-Figure 8 and Table 2: For RNA interference experiments, three experimental groups (dsSinvNPC2a, dsSinvNPC2b, dsEGFP control) were established. In these figures, what about that feeding and dsSinvNPC2b? What about that SinvNPC2b protein docking with EDMP? What about that ligand binding assays of SinvNPC2b with 1-NPN and EDMP at pH 7.4? And what about that binding affinities of SinvNPC2b for the tested ligands in Table 2?

Author Response

(The authors gave the same response as above.)

Reviewer 3 Report

Comments and Suggestions for Authors

The reviewer has read with interest the manuscript entitled as “Functional Analysis of Niemann-Pick Type C2 (NPC2) in Alarm Pheromone Recognition by the Red Imported Fire Ant, Solenopsis invicta (Formicidae: Solenopsis)” submitted by Peng Lin, Jiacheng Shen, Xinyi Jiang, Fenghao Liu and Youming Hou to the science Journal INSECTS. The authors investigated the features of Nieman-Pick Type C2 proteins of the red imported fire ant, Solenopsis invicta and identified two NPC2 proteins (SinvNPC2a and SinvNPC2b), which they examined with molecular biological, electrophysiological, and behavioral methods. They found that SinvNPC2a is strongly related to olfactory stimulus reception of alarm pheromone EDMP. The response to the alarm pheromone EDMP was reduced after feeding dsSinvNPC2a, because the SinvNPC2a protein was not normally synthesized. However, what did happen in the OR receptor protein of EDMP in this time ? When SinvNPC2a was not synthesized, was the OR receptor protein of EDMP also not synthesized or functionally blocked? The reviewer hopes that the authors explain the relation between SinvNPC2a and OR receptor protein to EDMP in the response ability to EDMP. The moving speed was reduced after the feeding dsSinvNPC2a. Was it attributed to the damage of locomotor control by CNS or deficiency of sensory input? Though the reviewer evaluates this study good with hardly a correction, the reviewer found some careless errors as follows.

Line 10                   Studies Research --> Research

Line 15                   ants --> ants.

Line 31                   as a group --> The reviewer does not understand this phrase.

Line 261                 Figure 2A--> There is no Figure 2A

Line 281                 SIdentification --> Identification

Line 365                 The characters in The Figure 6D is too small. Please use somewhat larger characters.

Author Response

(The authors gave the same response as above.)
